# The Impact of Hydroxytyrosol on the Metallomic-Profile in an Animal Model of Alzheimer’s Disease

**DOI:** 10.3390/ijms241914950

**Published:** 2023-10-06

**Authors:** Miguel Tabanez, Ilma R. Santos, Juliane M. Ikebara, Mariana L. M. Camargo, Bianca A. Dos Santos, Bruna M. Freire, Bruno L. Batista, Silvia H. Takada, Rosanna Squitti, Alexandre H. Kihara, Giselle Cerchiaro

**Affiliations:** 1Center for Natural Sciences and Humanities, Federal University of ABC, Santo André 09210-580, SP, Brazil; miguel.tabanez@ufabc.edu.br (M.T.); ilma.regina@ufabc.edu.br (I.R.S.); mariana.camargo@ufabc.edu.br (M.L.M.C.); bruna.freire@ufabc.edu.br (B.M.F.); bruno.lemos@ufabc.edu.br (B.L.B.); 2Metal Biochemistry and Oxidative Stress Laboratory, Center for Natural Sciences and Humanities, Federal University of ABC, Santo André 09210-580, SP, Brazil; 3Center for Mathematics, Computing and Cognition, Federal University of ABC, São Bernardo do Campo 09606-045, SP, Brazil; juliane.ikebara@ufabc.edu.br (J.M.I.); bia.neuro@gmail.com (B.A.D.S.); silvia.takata@ufabc.edu.br (S.H.T.); alexandre.kihara@ufabc.edu.br (A.H.K.); 4Department of Laboratory Science, Ospedale Isola Tiberina—Gemelli Isola, 00186 Rome, Italy; rosanna.squitti.fw@fbf.isola.it

**Keywords:** copper, zinc, Alzheimer’s disease, animal studies, hydroxytyrosol, metallomics

## Abstract

It is undeniable that as people get older, they become progressively more susceptible to neurodegenerative illnesses such as Alzheimer’s disease (AD). Memory loss is a prominent symptom of this condition and can be exacerbated by uneven levels of certain metals. This study used inductively coupled plasma mass spectrometry (ICP-MS) to examine the levels of metals in the blood plasma, frontal cortex, and hippocampus of Wistar rats with AD induced by streptozotocin (STZ). It also tested the effects of the antioxidant hydroxytyrosol (HT) on metal levels. The Barnes maze behavior test was used, and the STZ group showed less certainty and greater distance when exploring the Barnes maze than the control group. The results also indicated that the control group and the STZ + HT group exhibited enhanced learning curves during the Barnes maze training as compared to the STZ group. The ICP-MS analysis showed that the STZ group had lower levels of cobalt in their blood plasma than the control group, while the calcium levels in the frontal cortex of the STZ + HT group were higher than in the control group. The most important finding was that copper levels in the frontal cortex from STZ-treated animals were higher than in the control group, and that the STZ + HT group returned to equivalent levels to the control group. The antioxidant HT can restore copper levels to their basal physiological state. This finding may help explain HT’s potential beneficial effect in AD-patients.

## 1. Introduction 

Neurodegenerative diseases are disorders that specifically impact the functioning of neurons, possibly accumulating some protein [1]. The pathologies with the highest incidence are Alzheimer’s disease (AD), followed by Parkinson’s disease (PD), vascular and frontal temporal dementias, in addition to Huntington’s disease (HD) [2]. Alzheimer’s Disease International estimates 50 million people with dementia worldwide in 2020, reaching 82 million and 152 million in 2030 and 2050, respectively. Currently, 60% live in developing countries, which could increase to 71% by 2050 [3].

The clinical progression of AD is divided into four stages, determined by clinical symptoms and neuropathological analysis [4]. The cause of AD is currently unknown. However, its progression and diagnosis may be linked to the buildup of beta-amyloid proteins (Aβ) around cells and the folding of tau proteins inside them, which leads to increased phosphorylation. The accumulation of Aβ can cause issues with neuronal functioning, affecting synaptic functions, axonal transport, mitochondrial functions, and protein degradation through the ubiquitin system [5,6,7].

Studies have indicated that following the Mediterranean diet is associated with a lower prevalence of AD. This diet includes plenty of vegetables, legumes, fruits, cereals, unsaturated fat, moderate to high amounts of fish, limited red meat and saturated fat, and occasional alcohol consumption in moderation [8,9,10]. Olive oil is a key component of a Mediterranean diet and has been found to have neuroprotective properties due to its polyphenolic molecules. These molecules can combat free radicals, reduce inflammation, and prevent cell death. Additionally, olive oil’s chelating action can prevent DNA damage from iron and reduce the peroxidation of low-density lipids from copper [11,12,13,14,15]. The crucial ingredient in olive oil is hydroxytyrosol, which functions as an antioxidant and is linked with guarding effects against AD [16,17,18]: it improves mitochondrial functions and modulates microglia neuroinflammation in the brain cortex, also improving oxidative stress [19,20,21,22].

Metal ions are known to play a pivotal role in the proper function of the brain. However, patients who suffer from AD frequently exhibit imbalances in their basal metal ion concentrations, which can lead to metal dysfunction. Iron, copper, zinc, and calcium levels have also been found to be interrelated with AD progression. Clinical tests have explored the application of supplementation or chelators to modulate these levels. Additionally, manganese and cobalt concentrations may also be related to AD progression.

Researchers frequently employ the intracerebroventricular injection (icv) of the drug streptozotocin (STZ) to simulate AD in vivo models [23]. This drug (recommended for use only in male rats) is believed to initiate the AD disease cascade by reducing glucose utilization in the brain, leading to metabolic and behavioral disorders and a decrease in ATP and phosphocreatine concentrations [24]. Subsequently, the amyloid cascade occurs, in which higher concentrations of Aβ oligomers generate Aβ plaques, triggering tau hyperphosphorylation, eventually leading to neurodegeneration which resembles dementia [25,26,27].

This study aims to investigate: (i) the impact of neurodegeneration induced in vivo by STZ in 2-month-old male Wistar rats; (ii) the effect of the antioxidant hydroxytyrosol in recovering neurodegeneration induced by STZ; and (iii) the metallomic profile in this dynamic of neurodegeneration and recovery.

## 2. Results

### 2.1. Barnes Maze Test

We conducted a study using a modified Barnes maze procedure to evaluate the impact of icv-STZ and STZ + HT on spatial memory [28,29]. During the acquisition phase, the time to reach the escape box was reduced to four days. In Figure 1 it can be seen the elapsed escape time, meaning the time that the animal finds the target, comparing training days and trials by training. As observed, the control group (Figure 1A) and STZ + HT group (Figure 1C) exhibited a more significant difference in training abilities compared to the STZ-only group (Figure 1B), from the first to the fourth day of the trial. Figure 2 shows the results from distance followed by the animal: also, we can see the statistical difference in learning curves in the control (Figure 2A) and STZ + HT (Figure 2C) groups, and not in the STZ-only group (Figure 2B), meaning that STZ affected animals had the direction sense negatively affected and run the same distance in the first and fourth days of trials.

Figure 3 shows a parameter used to control if the stereotaxic surgery affects the animal velocity in the maze: the average speed. As we can see, there was no difference in all groups during the trials, meaning that the surgery did not affect animals’ body and movements.

Representing the final and most crucial test, in the probe trial where the escape box was removed (Figure 4), directly comparing the animal groups, STZ group showed a lower percentage of target exploration compared to the vehicle group (Vehicle: 24.22 ± 2.45%; STZ: 16.99 ± 1.88%; *p* = 0.0480; Figure 4C). Elapse time to target (STZ *p =* 0.16; STZ + HT *p =* 0.20, Figure 4A), time at target (STZ *p =* 0.15; STZ + HT *p =* 0.08, Figure 4B), quadrant exists (STZ *p =* 0.13; STZ + HT *p =* 0.12, Figure 4E) and distance (STZ *p =* 0.37; STZ + HT *p =* 0.28, Figure 4F) displayed trends in these parameters, no statistical difference was seen compared to vehicle group. No trend exists in the % quadrant time parameter, STZ and STZ + HT *p* > 0.9 (Figure 4D). So, HT did not improve memory impairment in STZ rats in the probe trial of Barnes maze. These results may suggest that the possible changes in metal ions in the brain caused by HT did not affect cognitive function.

### 2.2. Metal Content in Tissue Samples

After the Barnes maze probe trial, the animal was euthanized, blood samples were collected, and the rat’s corps were perfused with saline solution. After perfusion, the brain was collected, the frontal cortex and hippocampus were dissected and frozen, as well as the blood plasma. The metal content in brain areas is associated with pathology AD [30,31], and their levels in plasma are helpful to serve as non-invasive biomarkers., and plasma is helpful. We performed ICP-MS to examine the concentration of six metals in these tissues. Sample preparation was made in nitric acid with pre-digestion and digestion.

#### 2.2.1. Metallomics Profile of Plasma

We noticed only one statistical difference in the Metallomics profile from plasma by ICP-MS (Figure 5), whereas the STZ group displayed a lower concentration of cobalt compared to the vehicle group (Vehicle: 24.09 ± 3.61 µg/kg; STZ: 13.04 ± 1.29 µg/kg; *p* = 0.0357; Figure 5D). The STZ + HT group did not show a statistical difference (STZ + HT: 18.66 ± 5.94 µg/kg, *p* = 0.6484).

#### 2.2.2. Metallomics Profile of the Frontal Cortex

It was possible to detect two statistical differences (Figure 6) in Ca and Cu the study groups between. Ca displayed a higher concentration in the STZ + HT group compared to the vehicle group (vehicle: 384.50 ± 15.00 mg/kg; STZ + HT: 1572.2 ± 386.29 mg/kg; *p* = 0.0493; Figure 6A), whereas STZ group showed a trend without a statistic difference (STZ: 793.36 ± 286.68 mg/kg; *p* = 0.1909). STZ group displayed a higher concentration of copper in frontal cortex tissues than the vehicle group, whereas STZ + HT equals vehicle group (Vehicle: 9.62 ± 0.38 mg/kg; STZ: 11.87 ± 0.95 mg/kg; *p* = 0.0292; Figure 6E).

#### 2.2.3. Metallomics Profile of the Hippocampus

ICP-MS results displayed statistical differences in hippocampus tissue (Figure 7). Calcium concentration was higher in the hippocampus from the STZ group than in the vehicle group (Vehicle: 286.02 ± 25.17 mg/kg; STZ: 438.16 ± 49.09 mg/kg; one-way ANOVA *p* = 0.0185; Figure 7A). Such as calcium, iron also displayed a higher concentration in hippocampus tissues from the STZ group than the vehicle group (Vehicle: 47.12 ± 2.25 mg/kg; STZ: 66.05 ± 5.24 mg/kg; *p* = 0.0049; Figure 7C). STZ + HT group displayed statistically equal concentrations of calcium and iron compared to the vehicle group (*p* = 0.859 and *p* = 0.735, respectively). Cobalt analyses also showed a statistical difference in hippocampus tissue, the STZ + HT group exhibited lower concentrations of cobalt compared to the vehicle group (Vehicle: 22.51 ± 2.05 µg/kg; STZ + HT: 17.38 ± 1.13 µg/kg; *p* = 0.0049; Figure 7D). STZ group exhibited a trend with lower but no statistical difference (STZ: 20.03 ± 1.04; *p* = 0.40) [32,33].

## 3. Discussion

The best result of our study is that copper levels in the frontal cortex of STZ-treated animals were higher than in the control group and that the treatment with HT restored copper physiological levels. Moreover, in the animals treated with HT, the concentration of iron returned to its average level in the hippocampus, similar to the control group.

Other significant results were that the STZ group had lower levels of cobalt in their blood plasma than the control group, while the calcium levels in the frontal cortex of the STZ + HT group were higher than in the control group. The behavioral studies revealed that the STZ-treated animals only showed a difference in their target exploration during the final trial when compared to the control animals.

The current results show that STZ causes copper dysregulation in the frontal cortex. The dysregulation of zinc and manganese has been linked to the incidence and progression of AD [34,35]. However, upon analysis of various tissues, we detected no significant difference in the concentrations of zinc and manganese between the groups. It is well known that STZ can induce Type 1 diabetes models, which disrupt copper homeostasis, leading to increased copper levels in various tissues and organs after the first week of diabetes onset. This information has been reviewed in [36], and there is also an increase in urinary copper excretion, indicating an increase in the non-ceruloplasmin-Cu fraction in the blood [37]. Some authors have suggested that diabetes mellitus can hinder ATPase7B, leading to increased levels of nonceruloplasmin-Cu [38,39]. Our current findings support this idea, making the specific effect of HT in restoring copper physiology in a critical brain structure quite valuable. Additionally, recent research has revealed that copper dyshomeostasis in AD is directly linked to cortical-subcortical circuitry in the frontal cortex, where necessary copper enzymes such as dopamine beta-hydroxylase (DBH) may be involved, leading to a catecholamine imbalance [39]. Current results on the STZ AD model align with the reported evidence and with a recent clinical trial with a similar molecule in AD patients [40]. The most recent and inclusive meta-analysis of copper in AD specimens and serum/plasma samples is shown in the literature [41]. The Amyloid cascade hypothesis suggests that monoclonal antibodies (donanemab, lecanemab, and aducanumab) targeting Aβ plaques have demonstrated a certain level of promise in reducing cognitive decline in AD patients. Still, it is important to note that this is only by 27% (Lecanemab) to 35% (Donanemab, Trailblazer-Alz2 study Ely Lilly). This still leaves approximately 65% of the disease burden without any disease-modifying therapeutic options. It is worth highlighting that monoclonal antibodies against Aβ can have severe side effects (ARIA). Therefore, compounds such HT, olive polyphenols, or other metal-attenuating compounds could potentially serve as adjuvants in AD to offer some value. The most recent findings on the STZ AD model align with the reported evidence and a recent clinical trial involving a similar molecule for AD patients [41].

Regarding the behavioral study, the Barnes maze test was utilized to assess differences in spatial learning and memory between the groups. Neurodegeneration was induced by intracerebroventricular STZ injection, leading to a pathology resembling AD [23,24,42,43,44,45]. Other studies that evaluated the cognitive improvement action of HT in animals used daily gavage of HT acetate 50 mg/kg for 12 weeks [46] or 5 mg/kg/day for six months [47]. A specific administration method was developed to reduce the amount of HT required. The decision was made to administer HT at the same dosage as previously established by Luccarini et al., which is 450 µmol.L^−1^ [48]. In order to prevent oxidation reactions, the HT was dissolved in ultra-pure water, as STZ is dissolved in acid citrate buffer. The injection of HT occurred one hour before the STZ injection.

The Barnes maze has been utilized as an assessment tool for evaluating spatial learning abilities and memory in animals [49,50]. Previous studies have investigated the effects of STZ at various doses and over different periods in the Barnes maze [45,51]. Throughout the training phase, we observed a notable difference in learning curves between the control and STZ + HT groups (from the first to the fourth day of training): the STZ group traveled a greater distance than the vehicle group to attain the target (and did not learn to search the target). So, the STZ + HT group demonstrated comparable performance to the vehicle group, indicating that HT facilitates learning speed retention. During the final examination, the sole significant difference between groups was the percentage of target exploration, with the STZ group exploring the target hole less than the vehicle group, which aligns with the prior literature findings [51].

The Metallomics profiles of animal tissues were analyzed via ICP-MS, following a meticulous process of weighing, pre-digestion, and acid digestion. Specifically, the hippocampus, frontal cortex, and blood plasma tissues were subjected to acid digestion using hot nitric acid before being diluted with ultrapure water. Our findings indicate that the blood plasma of the STZ group exhibited a lower concentration of cobalt in comparison to the vehicle group. However, it is worth noting that the STZ + HT group did not display such a difference, which suggests the effectiveness of HT in preventing chemical disturbances. It is crucial to highlight that cobalt is closely linked to vitamin B12 (cobalamin), which is involved in iron metabolism and in red blood cell biosynthesis. Low cobalamin concentrations have been found in 10–25% of older adults, associated with a higher risk of AD. Nevertheless, it is interesting to note that patients diagnosed with AD do not exhibit lower concentrations of cobalt in their blood plasma [52,53,54,55] but they have iron disturbances [56]. The lower cobalt concentration in blood plasma caused by STZ demonstrates its efficiency in interfering with the cobalamin transport and absorption cycle.

After analyzing the frontal cortex, the Metallomics study revealed two significant deviations from the vehicle group. The STZ + HT group demonstrated increased calcium concentration in this tissue, which has been previously reported in the literature following STZ injection. However, the kidneys and liver were the organs analyzed after injection into the renal artery, and a high calcium concentration in these tissues resulted in lipid peroxidation, inflammatory response, and disruption of the redox-nitroso balance, ultimately leading to cellular damage [57]. It was observed that the previous administration of HT had a negative impact on the calcium levels, causing an increase in concentration. Studies conducted by Palmerini et al. have shown that HT has a similar effect on human lymphocytes, which might be due to an increase in calcium absorption or hindrance in calcium release from cellular stores. Another difference was noted in the copper levels [58]. The STZ group had higher copper concentration when compared to the vehicle group. However, the STZ + HT group managed to return to the normal physiological copper concentration in the tissue. A meta-analysis by Schrag et al. reported that AD patients had lower copper concentrations in the brain. Nevertheless, higher levels of free copper can produce reactive oxygen species, causing damage to the cells [59,60,61]. Upon comparing the concentrations of copper in a sample exposed to STZ to that of a control group, it becomes evident that STZ is capable of inducing damage through the increase of free copper. However, it remains challenging to ascertain whether HT positively affects STZ’s impact on the cortex. This is due to the fact that, even if HT were to restore copper levels to normal, the damage incurred as a result of increased calcium concentration may still be of significant concern.

Metallomics analysis in the hippocampus indicates significant changes in the concentration of three crucial elements: calcium, iron, and cobalt. The STZ group exhibited higher calcium levels compared to the vehicle group, while the STZ + HT group demonstrated a return to baseline calcium concentration, which was not observed in the frontal cortex. It is evident that HT has distinct effects on the hippocampus and frontal cortex tissue. Recent research has indicated that HT plays a vital role in reducing oxidative stress and neuroinflammation in mouse models with depression. Additionally, pregnant sows treated with 1.5 mg HT per day displayed a notable increase in the concentration of neurotransmitters such as noradrenaline, 3,4-dihydroxyphenylacetic acid, homovanillic acid, and serotonin in the fetal pig’s hippocampus [62,63]. There was a higher iron concentration in the group treated with STZ than in the control group. However, in the STZ + HT group, the concentration of iron returned to its normal level, similar to the control group. This finding aligns with previous studies on patients with AD, who also show higher concentrations of iron in the hippocampus and other regions of the brain, including the deep gray matter and neocortical region [64,65]. Just like copper, increased levels of iron can cause an increase in reactive oxygen species and ferritin, while lowering levels of ferroportin. These alterations may pose a threat to the structural stability of the hippocampus [66,67]. The pre-treatment of HT and STZ injection results showed significantly reduced cobalt concentrations compared to the control group. Cobalt has the ability to inhibit or inactivate the Pin1 protein, which is a key player in stress response, cell growth, immune function, and germ cells. Pin1 plays a vital role in preventing neurodegeneration by accelerating the dephosphorylation process of phosphorylated tau and APP [63]. With the previous administration of HT injections, there is a plausible chance of reporting a positive outcome.

## 4. Materials and Methods

### 4.1. Animals and Housing

Male Wistar rats (2 months old; weight 255–380 g) were obtained from ICB-USP (São Paulo, SP, Brazil) and housed in white PVC (40 × 33 × 17 cm) in a room with controlled temperature (24 °C), 40–55% relative humidity, 12/12 light–dark cycle and free access to fresh water and food. All procedures were performed through the morning period and approved by and are following the local ethics committee from the Universidade Federal do ABC (CEUA/UFABC, protocol number 5751061119).

### 4.2. Surgery and Injection Procedure

Rats were anesthetized with Isoflurane (4% induction, 1–2% maintenance with 0.8 mmHg O_2_). The top of the animal’s head was shaved and cleaned with polyvidone iodine solution. Lidocaine was injected at the skull top, and after 5 min, an incision (3–4 cm) was made with a scalpel. The skull’s top was cleaned with peroxide hydrogen. The cannula was fixed with the stereotaxic coordinates for unilateral intracerebroventricular (ICV) injection, 0 mm posterior from bregma, 1.4 mm lateral to sagittal suture, and 3.5 mm beneath brain surface [68]. Animals received meloxicam (2 mg/kg), enrofloxacin (10 mg/kg), and tramadol (7.5 mg/kg). After the surgery procedure, animals were caged and rested for 5 days.

The rats were separated into three distinct groups for the experiment and rested 5 to 7 days before the surgery. The sample size of the groups was calculated following published procedures [69]. They were given two rounds of injections—the first on day one and the second on day three. Vehicle rats received 4 µL of 0.05 mol/L citrate buffer, pH 4.5; STZ group received 4 µL of 3 mg/kg STZ (Sigma, St. Louis, MO, USA) freshly dissolved into citrate buffer [42,45]; STZ + HT group received 1 µL of 450 µmol·L^−1^ of hydroxytyrosol, after 1 h, animals received 3 µL of 3 mg/kg [48,70]. The animals were kept in cages and allowed to rest until the Barnes maze behavior test day.

### 4.3. Barnes Maze Behavior Test

The protocol used for the Barnes maze experiment was adapted from three sources: [42,50,71]. The experiment was performed the fourth week after injection on a circular platform (120 cm diameter) with 20 peripheral holes (10 cm diameter) evenly spaced around the perimeter. In the experimental room, visual clues were located on every wall. A dark escape box was positioned under the target hole. The acquisition phase took place over four days, with four trials per day and a minimum 15-min intertrial interval. Before each trial, the animals were placed inside a dark cylinder on the platform for 30 s. The stopwatch was initiated when the cylinder was removed. The platform was cleaned with alcohol before each trial. During each trial, the animals had 3 min to explore the holes and platform while an aversive sound was playing (1550 Hz, 80–90 dB). If the animals entered the escape box, the sound was stopped. If the rat did not enter the box within 3 min, it was led gently to the target hole. On the fifth day, a probe trial was performed where the escape box was removed from the platform, and the maximum exploration time was reduced to 90 s. In the acquisition phase, escape latency, roamed distance, and average speed were analyzed. In the probe trial, quadrant exists, percent of quadrant time, percent of target exploration, elapsed time to target, time spent at target, and distance were counted. Videos were recorded using EthoVision XT (Noldus, Leesburg, VA, USA) and analyzed with Kinovea Software version 9.5. There were no deaths after the surgery, and the animals’ weights remained stable. During the intracerebroventricular injections (icv) procedure, the animals were anesthetized to minimize stress. Any wounds that occurred after icv were treated with veterinary care.

### 4.4. Blood Plasma Sampling and Brain Dissection

After completing the probe trial in the Barnes maze, the animals were administered urethane 3 g/kg i.p. dissolved in PBS buffer. After anesthetized, blood samples were collected with a syringe (previously washed with heparin 10 IU/mL) after a slight cut at the right atrium from the heart. Blood collection tubes were centrifuged (2000 rpm, 10 min), and blood plasma was collected. During blood samples centrifugation, animals were perfused with saline solution (0.9% NaCl), then animals were decapitated and brain collected, where the frontal cortex and hippocampus were dissection and frozen.

### 4.5. Metallomic Profile with ICP-MS

All tissues were lyophilized and then weighed in conical tubes (15 mL). Predigestion occurred through 48 h at room temperature with HNO_3_ proportionally to tissue weight. Subsequently, digestion occurred through 2 h at 90 °C with a water bath, and then ultrapure water was added to the final volume. Throughout this method, the conical tubes were weighed for dilution factor calculation. Ge and Rh (10 µg·L^−1^) were used as internal standards, also reference standards from bovine kidney [32] and water NIST 1640 were used for quality control. Calibration curves were made by stock solution in 5% *v*/*v* HNO_3_ at 1–200 µg/kg range. The concentrations of ^44^Ca, ^55^Mn, ^56^Fe, ^59^Co, ^65^Cu, and ^68^Zn were determined with inductively coupled plasma mass spectrometry (ICP-MS 7900, Agilent, Hachioji, Japan). Ca and Fe were analyzed with a high-energy helium collision chamber. Detection limits (LOD) were the following: 55.86 µg/L for Ca; 0.0383 µg/L for Mn; 2.167 µg/L for Fe; 0.0311 for µg/L Co; 0.481 µg/L for Cu and 2.159 µg/L for Zn.

The experimental design of this study is summarized in Figure 8.

### 4.6. Statistical Analysis

All results were analyzed by GraphPad Prism 8.0.1 software (San Diego, CA, USA), presented graphically as mean ± SEM and 95% confidence interval. The experiments of the training phase in the Barnes maze were analyzed by ANOVA—Two Ways or mixed-effects, with Bonferroni’s multiple comparison test. The ICP-MS analyses of animal material and the Barnes labyrinth probe test were analyzed by ANOVA—One Way or mixed-effects with Dunnett’s multiple comparison test.

## 5. Conclusions

The study conclusively demonstrates that intracerebroventricular injections of STZ can indeed modify the metallomics profile of animals, thereby supporting observations made in Alzheimer’s disease patients. This model can be effectively used to gain crucial insights into the disease’s progression. HT injection was somewhat effective in the Barnes maze test. Overall, the STZ + HT group displayed more similarities to the control group than the STZ group, making it an essential tool in AD research. It is important to highlight that HT has the ability to restore normal levels of copper in the frontal cortex and iron in the hippocampus, which is undoubtedly a valuable benefit. However, it cannot be ignored that this treatment threatens the cells due to the increased calcium levels in the frontal cortex. It is crucial to consider the possibility of using HT in future studies as a more cost-effective and efficient alternative to gavage for reducing the impact of STZ on other AD markers.

## Figures and Tables

**Figure 1 ijms-24-14950-f001:**
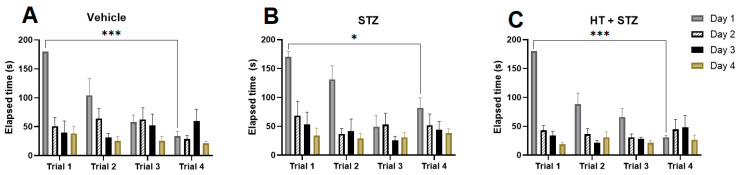
Elapsed escape time for target hole in Barnes maze during 4 training days. Representative data mean ± SEM (Vehicle n = 8, STZ n = 8, STZ + HT n = 9). * *p* ≤ 0.05 *** *p* ≤ 0.001. (**A**–**C**): analysis from the same group, divided by day trial. (**A**): Vehicle/control group (icv citrate injection); (**B**): STZ group (icv streptozotocin injection); (**C**): HT + STZ group (icv streptozotocin and hydroxytyrosol injections).

**Figure 2 ijms-24-14950-f002:**
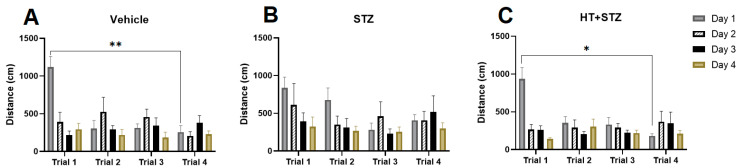
Distance traveled to the target hole in Barnes’ maze during 4 training days. Representative data of mean ± SEM (Vehicle n = 8, STZ n = 8, STZ + HT n = 9) and significant significance (ANOVA—Two Way or Mixed Model; Bonferroni Test). * *p* ≤ 0.05 ** *p* ≤ 0.005. (**A**–**C**): analysis from the same group, divided by day trial. (**A**): Vehicle/control group (icv citrate injection); (**B**): STZ group (icv streptozotocin injection); (**C**): HT + STZ group (icv streptozotocin and hydroxytyrosol injections).

**Figure 3 ijms-24-14950-f003:**
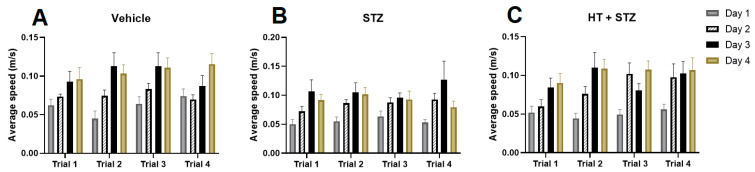
Average speed in Barnes’ maze during 4 training days. Representative data mean ± SEM (Vehicle n = 8, STZ n = 8, STZ + HT n = 9). There are no statistical differences. (**A**): Vehicle/control group (icv citrate injection); (**B**): STZ group (icv streptozotocin injection); (**C**): HT + STZ group (icv streptozotocin and hydroxytyrosol injections).

**Figure 4 ijms-24-14950-f004:**
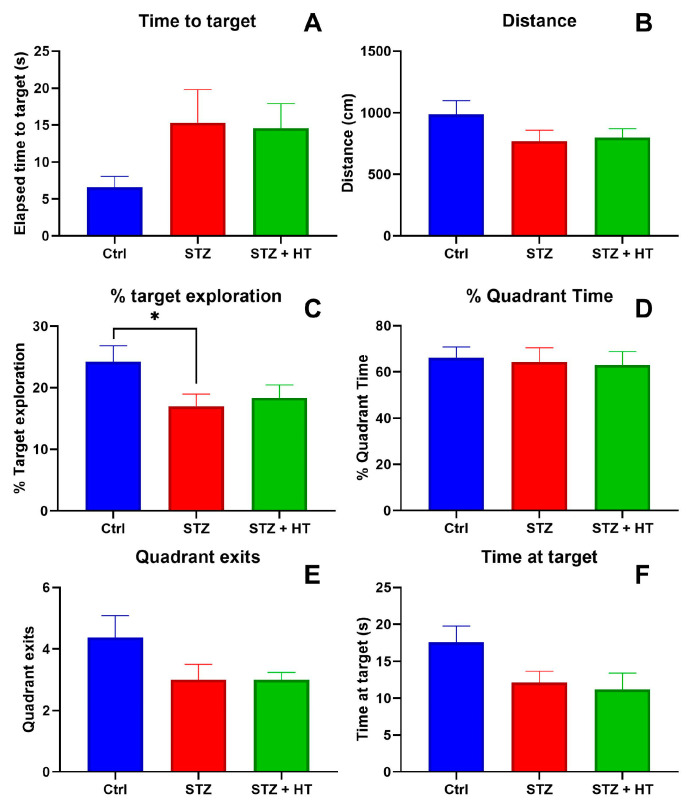
Parameters of final probe trial from Barnes maze (in the 5th day), comparing the experimental groups. (**A**) Time elapsed to target hole. (**B**) Total distance traveled. (**C**) Percentage of exploration in the correct hole. (**D**) Percentage of time in correct quadrant (Vehicle n = 7, STZ n = 8, STZ + HT = 9). (**E**) Number of outputs in the correct quadrant. (**F**) Time spent in target hole (Vehicle n = 8, STZ n = 8, STZ + HT = 9). Representative data of mean ± SEM and significant significance (ANOVA—One Way or Mixed Model; Dunnett Test) * *p* ≤ 0.05.

**Figure 5 ijms-24-14950-f005:**
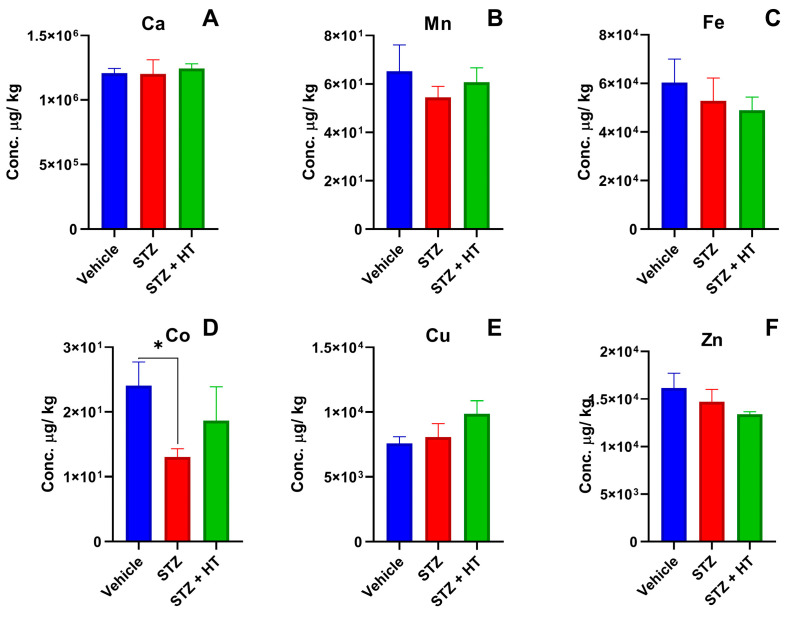
Concentration of metals from lyophilized blood plasma, being (**A**) calcium, (**B**) manganese, (**C**) iron, (**D**) cobalt, (**E**) copper, and (**F**) zinc. For Ca, Co, Cu, and Zn (Vehicle n = 8, STZ n = 8, STZ + HT = 9); for Mn (Vehicle n = 7, STZ n = 8, STZ + HT = 9) and for Fe (Vehicle n = 8, STZ n = 5, STZ + HT = 9). Representative data of mean ± SEM and significant significance (ANOVA—One Way or Mixed Model; Dunnett Test) * *p* ≤ 0.05.

**Figure 6 ijms-24-14950-f006:**
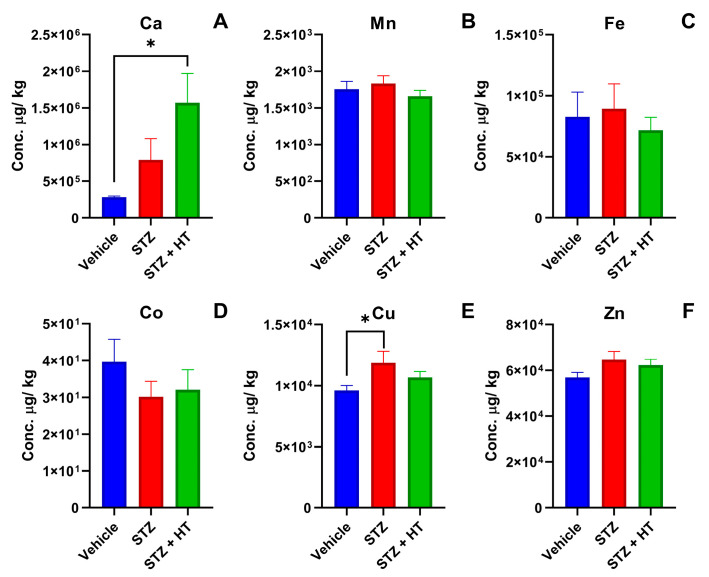
Concentration of metals from lyophilized frontal cortex tissue, being (**A**) calcium, (**B**) manganese, (**C**) iron, (**D**) cobalt, (**E**) copper, and (**F**) zinc. Representative data of mean ± SEM (Vehicle n = 8, STZ n = 8, STZ + HT = 9) and significant significance (ANOVA—One Way or Mixed Model; Dunnett Test) * *p* ≤ 0.05.

**Figure 7 ijms-24-14950-f007:**
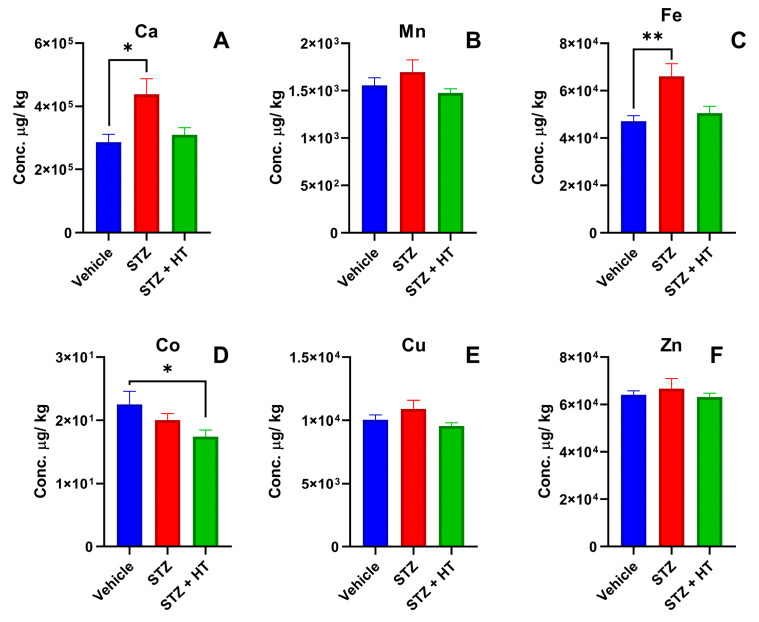
Concentration of metals from the lyophilized hippocampus, being (**A**) calcium, (**B**) manganese, (**C**) iron, (**D**) cobalt, (**E**) copper, and (**F**) zinc. Representative data of mean ± SEM (Vehicle n = 8, STZ n = 8, STZ + HT = 9) and significant significance (ANOVA—One Way or Mixed Model; Dunnett Test) * *p* ≤ 0.05 ** *p* ≤ 0.01.

**Figure 8 ijms-24-14950-f008:**
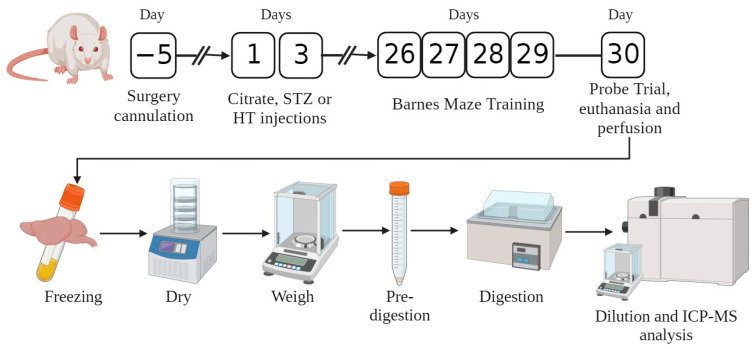
Schematic representation of the experimental design employed in this study.

## Data Availability

Experimental data or Barnes Maze trial videos can be provided upon request.

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
