# Peer review of "The Impact of Hydroxytyrosol on the Metallomic-Profile in an Animal Model of Alzheimer’s Disease"

_ijms, 2023, doi:10.3390/ijms241914950_

Round 1
Reviewer 1 Report
The manuscript submitted by Tabanez et al. describes the effects of the antioxidant hydroxytyrosol (HT) in the streptozotocin (STZ)-treated rat model of Alzheimer’s disease (AD). The authors’ goal was to determine the metallomic profile in this model and the effect of the antioxidant. They found tissue-specific metal imbalances in the STZ model, some of which were rescued by HT, and minor effects on behavior, one of which was rescued by HT. While potentially providing important insights, a number of issues must be addressed before the manuscript is suitable for publication. General and specific comments follow below.
General Comments:
The data presented is relevant to the field and the manuscript is generally well-structured, starting with the Barnes Maze results and then providing metallomics data. However, the manuscript would be improved by providing more data validating their model (i.e. AD pathology after STZ ± HT). The manuscript would also benefit from more background/context for metal dysregulation. There is no information provided about what is known in the literature, why these metals and regions were selected, and how their data fits within the field/published work.
Just over half of the cited references are more than 5 years old. While some of the original references are good to include (eg. the original description of the Barnes maze), there are other, more recent references that could be used for a more updated understanding. Many of the references are appropriate, and are included, but not sufficiently referenced/mentioned in the text. Furthermore, the list is inconsistently formatted.
While the experimental design appears to be appropriate to test the hypothesis, there are details and controls missing that make it hard to know for sure, as well as to know whether the results are reproducible and to draw meaningful conclusions. Specifically,
· Only male rats are used with no justification. This is significant for rigor and reproducibility, particularly in an AD model where the disease is known to exhibit sex differences.
· There is no mention of power analysis or reference to sample sizes based on prior published studies.
· Was there time for the rats to acclimate between their procurement and surgery/injection?
· There is no group of rats treated only with HT.
· There is no justification/rationale for the HT dose and how that compares to what has been previously used.
The graphs properly show the data, they are easy to interpret and understand. The data is generally interpreted correctly, however, the authors at times overstate their results/conclusions.
The ethics and data availability statements are adequate.
Specific Comments:
The introduction paragraph about metal ion imbalance that begins on line 55 is missing the necessary references and details about which tissues/brain regions exhibit these imbalances.
The introduction is also missing information about hydroxytyrosol (HT). There is no explicit reasoning for why it was studied, what the translational relevance is, etc.
Section 2.2 of the results section addresses metal content in various tissue samples. There is no justification or rationale presented for why the authors chose to examine plasma, cortical, and hippocampal tissues. Nor was there any justification/rationale for the six metals they presented. Did they look at others and not present them, or did they restrict their analysis to these six, and if so, why?
There is also no clear statement about what the expected effect of STZ, or HT, on metals would be.
Please comment on the HT dose selected (why this dose, its relevance to the human condition) and whether the effects, or lack thereof, are due to the dose selected.
For some metals, the magnitude of the changes (due to STZ ± HT) is not that large. Do the authors consider them large enough to make a meaningful difference?
To start the discussion section (line 222), the authors state “The best result of our study is that copper levels in the frontal cortex of STZ-treated animals were higher than in the control group, and that the treatment with HT restored copper physiological levels.” While this is a true statement about STZ-treated mice compared to the vehicle controls, STZ+HT is not significantly different from either STZ or vehicle control, and in fact appears to be midway between them, perhaps showing a partial rescue. Significantly, this relates to the above points regarding HT dose and meaningful magnitudes of change.
Lines 232-234 mention the lack of change in zinc and manganese. Please comment on why this would be the case and its significance.
Lines 259-268 discuss the results of the Barnes maze. As there seemed to be little effect of STZ on many of the study’s outcomes, validation of the STZ treatment paradigm are necessary. As it stands, its hard to know whether the lack of STZ effects are because they are truly absent, or the STZ treatment didn’t work. Additional experiments to verify the treatment induced an AD-like phenotype and referencing prior studies from the literature are necessary to then draw meaningful conclusions from this work.
Please discuss why there would be consistent changes in cobalt across all tissues examined, even if they were not significant in the cortex. Why would Co be singularly affected in this way? Is this unique to the STZ model or AD models more generally?
Please comment on why HT rescues Ca, Mn, Fe, Cu, and Zn, but would exacerbate the STZ-induced decrease in Co. Why would this happen, and how?
Please comment on why there appear to be different effects of STZ based on the metal and the tissue.
The conclusion section (lines 435-448) is generally overstated for the points made above. Please temper this section to more accurately reflect the modest changes observed.
There are a few typographical and grammatical errors that need to be corrected, but for the most part, the quality of the English language is fine.
Author Response
Comments and Suggestions for Authors
The manuscript submitted by Tabanez et al. describes the effects of the antioxidant hydroxytyrosol (HT) in the streptozotocin (STZ)-treated rat model of Alzheimer’s disease (AD). The authors’ goal was to determine the metallomic profile in this model and the effect of the antioxidant. They found tissue-specific metal imbalances in the STZ model, some of which were rescued by HT, and minor effects on behavior, one of which was rescued by HT. While potentially providing important insights, a number of issues must be addressed before the manuscript is suitable for publication. General and specific comments follow below.
General Comments:
The data presented is relevant to the field and the manuscript is generally well-structured, starting with the Barnes Maze results and then providing metallomics data. However, the manuscript would be improved by providing more data validating their model (i.e. AD pathology after STZ ± HT). The manuscript would also benefit from more background/context for metal dysregulation. There is no information provided about what is known in the literature, why these metals and regions were selected, and how their data fits within the field/published work.
RESPONSE: First, we would like to thank the referee for carefully reading this work and providing valuable comments that have certainly improved it.
We provided the literature information and included it in the text, as well as new references (in yellow in the corrected manuscript).
Just over half of the cited references are more than 5 years old. While some of the original references are good to include (eg. the original description of the Barnes maze), there are other, more recent references that could be used for a more updated understanding. Many of the references are appropriate, and are included, but not sufficiently referenced/mentioned in the text. Furthermore, the list is inconsistently formatted.
RESPONSE: We formatted the list of references, and provided the new references.
While the experimental design appears to be appropriate to test the hypothesis, there are details and controls missing that make it hard to know for sure, as well as to know whether the results are reproducible and to draw meaningful conclusions. Specifically,
- Only male rats are used with no justification. This is significant for rigor and reproducibility, particularly in an AD model where the disease is known to exhibit sex differences.
RESPONSE: We understand the concern, but in this case, we can only use male rats in this method. Based on the method used by Carrettiero et al (Bioprotocol, 2019), they recommend:
- Here we recommend male rats since female rats present less sensitivity to STZ-induced cognitive impairment compared to male rats (Bao et al., 2017). Moreover, female rats display differences in some AD typical biomarkers compared to those observed in human. However, these differences were observed in Sprague-Dawley rats. More studies testing the sex differences related to AD susceptibility and progression in other rodent species and strains are needed to better understand how the hormonal modulation could affect the disease progression. As AD is more prevalent between women than men, the fact that STZ models are generally applied only to male rodents is a bias that restricts the representativeness of the model (Bao et al., 2017).
So, to use correctly this method to generate spontaneous AD, we follow the references from Carrettiero and Bao et al. We think that it may need one work only to provide sex differences in this model, and this was not the objective in this work.
- There is no mention of power analysis or reference to sample sizes based on prior published studies.
RESPONSE: Also, we followed the method from Carrettiero et al, and we calculated the sample size from the formula from:
Suchal, K., Malik, S., Khan, S. I., Malhotra, R. K., Goyal, S. N., Bhatia, J., Arya, D. S. (2017). Protective effect of mangiferin on myocardial ischemia-reperfusion injury in streptozotocin-induced diabetic rats: role of AGE-RAGE/MAPK pathways. Scientific Reports, 7(February), 42027. http://doi.org/10.1038/srep42027
(the reference was inserted in the manuscript in the experimental part)
using the formula n= 1 + [2C*(s/d)2] considering p<0.05 significant, test power of 90%, maximum deviation of 20% and expected difference between groups of 50%, the value of n=5 animals per group was found.
We use 8 -9 animals per group, even more than the minimum calculated.
*Explaining the formula:
where
C is dependent on the values chosen for the strength or power of the test (1-p; chance of finding an existing difference) and level of significance (p; the chance of considering two groups different when they are not). For researchers who consider p<0.05 the value of p is 0.05 is the acceptable standard deviation according to the researcher's projection and d is the expected difference between the groups.
To calculate C, the formula must be applied:
C= (zp + zp)2
Where z corresponds to values found in statistics books and at the bottom of this page. It is important to remember that to determine the z, a value you must divide the confidence interval value by 2, for example, for an interval of 0.95 the value to look for in the table is 0.475 and the sum of the values in the row (1.9) with the top of the column (0.06) gives you a value of z=1.96. In experiments in the area of health, the power of the test is commonly 90% for which the z value is 1.282. For reference see the article Suchal et al.
Therefore, if in a given experimental model, a researcher wants to work with a power for the test of 90% and a level of significance 0.05, the value of C will be (1.96 + 1.282)2=10.51
Considering also a maximum deviation (s) of 0.2 (20%) and an expected difference between groups (d) of 0.5 (50%), when applying the formula
n=1+[2*10.51*(0.2/0.5)2]
The result will be an n of 4.36 animals, which rounded to the next whole number will be 5 animals per group.
- Was there time for the rats to acclimate between their procurement and surgery/injection?
RESPONSE: The time was from 5 to 7 days. Included in the experimental part.
- There is no group of rats treated only with HT.
RESPONSE: In this study, we did not treat only HT as previously we found that HT is not toxic to Hippocampus cell culture (Hippo E2, cerdallane, Canada), even in high concentrations used.
- There is no justification/rationale for the HT dose and how that compares to what has been previously used.
RESPONSE: We need to chose one type of inoculation (to see the direct effect we chose icv injections) and dose for this first study, and the dose was based on and adapted from the paper (already cited in experimental section): Luccarini I, Ed Dami T, Grossi C, Rigacci S, Stefani M, Casamenti F. Oleuropein aglycone counteracts Aβ42 tox-icity in the rat brain. Neurosci Lett 2014;558:67–72. Available from: http://dx.doi.org/10.1016/j.neulet.2013.10.062
In fact there was an explanation in discussion:
Regarding the behavioral study, the Barnes maze test was utilized to assess differ-ences in spatial learning and memory between the groups. Neurodegeneration was in-duced by intracerebroventricular STZ injection, leading to a pathology resembling AD (42–45). Other studies that evaluated the cognitive improvement action of HT in animals used daily gavage of HT acetate 50 mg/kg for 12 weeks (46) or 5 mg/kg/day for six months (47). A specific administration method was developed to reduce the amount of HT re-quired. The decision was made to administer HT at the same dosage as previously established by Luccarini et al., which is 450 µmol.L-1(48). In order to prevent oxidation reactions, the HT was dissolved in ultra-pure water, as STZ is dissolved in acid citrate buffer. The injection of HT occurred one hour before the STZ injection.
The graphs properly show the data, they are easy to interpret and understand. The data is generally interpreted correctly, however, the authors at times overstate their results/conclusions.
The ethics and data availability statements are adequate.
Specific Comments:
The introduction paragraph about metal ion imbalance that begins on line 55 is missing the necessary references and details about which tissues/brain regions exhibit these imbalances.
RESPONSE: We included in the introduction the references missing.
The introduction is also missing information about hydroxytyrosol (HT). There is no explicit reasoning for why it was studied, what the translational relevance is, etc.
RESPONSE: We included in the introduction this part and references.
Section 2.2 of the results section addresses metal content in various tissue samples. There is no justification or rationale presented for why the authors chose to examine plasma, cortical, and hippocampal tissues. Nor was there any justification/rationale for the six metals they presented. Did they look at others and not present them, or did they restrict their analysis to these six, and if so, why?
RESPONSE: We included in the introduction this part, with references.
There is also no clear statement about what the expected effect of STZ, or HT, on metals would be.
RESPONSE: We expected some changes, as it occurs.
Please comment on the HT dose selected (why this dose, its relevance to the human condition) and whether the effects, or lack thereof, are due to the dose selected.
RESPONSE: The dose was adapted from previous studies (cited in the previous response) and we adapted for this study, with icv injections.
For some metals, the magnitude of the changes (due to STZ ± HT) is not that large. Do the authors consider them large enough to make a meaningful difference?
RESPONSE: We consider only changes with statistical significance.
To start the discussion section (line 222), the authors state “The best result of our study is that copper levels in the frontal cortex of STZ-treated animals were higher than in the control group, and that the treatment with HT restored copper physiological levels.” While this is a true statement about STZ-treated mice compared to the vehicle controls, STZ+HT is not significantly different from either STZ or vehicle control, and in fact appears to be midway between them, perhaps showing a partial rescue. Significantly, this relates to the above points regarding HT dose and meaningful magnitudes of change.
RESPONSE: We consider for the next work to change the dose, and also change the method to include HT in the diet (gavage for example). This work was the first, and the objective is to show changes (or not) in metal content after the icv injections.
Lines 232-234 mention the lack of change in zinc and manganese. Please comment on why this would be the case and its significance.
RESPONSE: As we can expect differences because of human literature in this way, we did not find them. So we are not able to discuss about it, why we cannot find differences. Maybe it is because AD did not cause so many changes in Zn and Mn in the brain, but we chose not to hypothitize this.
Lines 259-268 discuss the results of the Barnes maze. As there seemed to be little effect of STZ on many of the study’s outcomes, validation of the STZ treatment paradigm are necessary. As it stands, its hard to know whether the lack of STZ effects are because they are truly absent, or the STZ treatment didn’t work. Additional experiments to verify the treatment induced an AD-like phenotype and referencing prior studies from the literature are necessary to then draw meaningful conclusions from this work.
RESPONSE: Effectively the Barnes maze is not so sensitive as well as aquatic maze for rats. However, based on solid literature, we used STZ to cause AD in rats (literature cited in the manuscript).
We included more analysis, separating the tests by DAY, not only by TRIAL, and we can see more differences now (New Figures 1-3), as the learning curves. In fact the control and STZ+HT groups presented significative differences in the learning curves during the training. Instead, the STZ did not present the learning curve from the first from forth day of trial. With these data – separated for DAYS of trials in the same group, it was possible to analyze and see more differences than comparing only the groups (as it was before).
Please discuss why there would be consistent changes in cobalt across all tissues examined, even if they were not significant in the cortex. Why would Co be singularly affected in this way? Is this unique to the STZ model or AD models more generally?
RESPONSE: effectively this would be a new and long study, which we studied the possibility of carrying out, but not to include in this work, in this work we detail the effects, but we did not carry out protein tracking involving each metal to verify the effects on each one of them, if they are the ones affected and so on. This would be a long and upcoming work.
Please comment on why HT rescues Ca, Mn, Fe, Cu, and Zn, but would exacerbate the STZ-induced decrease in Co. Why would this happen, and how?
RESPONSE: effectively this would be a new and long study, which we studied the possibility of carrying out, but not to include in this work, in this work we detail the effects, but we did not carry out protein tracking involving each metal to verify the effects on each one of them, if they are the ones affected and so on. This would be a long and upcoming work.
Please comment on why there appear to be different effects of STZ based on the metal and the tissue.
RESPONSE: We can conclude that is because the methabolism of this drug, but we are not sure, as we cannot find in literature its pharmacokinetics study.
The conclusion section (lines 435-448) is generally overstated for the points made above. Please temper this section to more accurately reflect the modest changes observed.
RESPONSE: We changed some statements.
Comments on the Quality of English Language
There are a few typographical and grammatical errors that need to be corrected, but for the most part, the quality of the English language is fine.
Reviewer 2 Report
In this manuscript, the authors examined the levels of metals in the blood plasma, frontal cortex, and hippocampus of streptozotocin (STZ)-induced AD model rats. The effects of the antioxidant hydroxytyrosol in STZ rats was also examined. Although there are some interesting results, I have following concerns.
1. In the Introduction, page 2, line 55-67, please cite references.
2. In the Introduction, the authors should describe the reason why they chose hydroxytyrosol.
3. The experimental design is a little confusing. I suggest the authors to make a figure showing the experimental schedule in detail.
4. Figure 1-3 are not correct. Figure 1 should be Escape time. Figure 2 should be Distance traveled. Figure 3 should be Average speed.
5. In Figure, 1 and 2, the authors should show the average escape latency and distance traveled of 4 trials each day, and compare the learning curve throughout the training from day1 to day4 by two-way ANOVA.
6. What about the effects of hydroxytyrosol on brain amyloid beta levels, a decrease in insulin signaling in the brain, etc. in STZ rats?
7. Hydroxytyrosol did not improve memory impairment in STZ rats in the probe trial of Barnes Maze as shown in Figure 4. This result suggests that the changes in metal ions in the brain by hydroxytyrosol do not affect cognitive function. The authors should discuss this point.
Author Response
In this manuscript, the authors examined the levels of metals in the blood plasma, frontal cortex, and hippocampus of streptozotocin (STZ)-induced AD model rats. The effects of the antioxidant hydroxytyrosol in STZ rats was also examined. Although there are some interesting results, I have following concerns.
- In the Introduction, page 2, line 55-67, please cite references.
RESPONSE: First, we would like to express our gratitude to the referee for carefully reading this work and providing useful comments that have certainly improved it.
We provided the literature information and included it in the text and new references (in yellow in the corrected manuscript).
- In the Introduction, the authors should describe the reason why they chose hydroxytyrosol.
RESPONSE: This part was missing and we included in the introduction (highlighted in yellow).
- The experimental design is a little confusing. I suggest the authors to make a figure showing the experimental schedule in detail.
RESPONSE: The details should be found step by step in the papers from Carrettiero et al (2019), which explain in details.
Also, a new Figure was added in the experimental part.
- Figure 1-3 are not correct. Figure 1 should be Escape time. Figure 2 should be Distance traveled. Figure 3 should be Average speed.
RESPONSE: We corrected and done new clearly figures.
- In Figure, 1 and 2, the authors should show the average escape latency and distance traveled of 4 trials each day, and compare the learning curve throughout the training from day1 to day4 by two-way ANOVA.
RESPONSE: We included these data in figures 1-3, making new figures where we can see not the differences in learning curves.
We included more analysis, separating the tests by DAY, not only by TRIAL, and we can see more differences now (New Figures 1-3), as the learning curves. In fact the control and STZ+HT groups presented significative differences in the learning curves during the training. Instead, the STZ did not present the learning curve from the first from forth day of trial. With these data – separated for DAYS of trials in the same group, it was possible to analyze and see more differences than comparing only the groups (as it was before).
We are grateful to referee to point us these new way to see these data.
- What about the effects of hydroxytyrosol on brain amyloid beta levels, a decrease in insulin signaling in the brain, etc. in STZ rats?
RESPONSE: Effectively, this would require a new and extensive study, which we considered but decided not to include in this work.
- Hydroxytyrosol did not improve memory impairment in STZ rats in the probe trial of Barnes Maze as shown in Figure 4. This result suggests that the changes in metal ions in the brain by hydroxytyrosol do not affect cognitive function. The authors should discuss this point.
RESPONSE: We included this statement in the manuscript, as it can be true.
Round 2
Reviewer 1 Report
The reviewer thanks the authors for their revisions and deems the manuscript suitable for publication.
The quality of the English is fine. There are a few places where the grammar could be improved, but the meaning is clear.
Reviewer 2 Report
The manuscript is significantly improved.